# The First Case of Human Hepatic Fasciolosis Presented as Hepatic Pseudotumor Histopathologically Diagnosed in Romania—A Case Report

**DOI:** 10.3390/healthcare11101451

**Published:** 2023-05-17

**Authors:** Victoria Birlutiu, Rares-Mircea Birlutiu

**Affiliations:** 1County Clinical Emergency Hospital Sibiu—Infectious Diseases Clinic, Faculty of Medicine Sibiu, Lucian Blaga University of Sibiu, Strada Lucian Blaga, Nr. 2A, 550169 Sibiu, Romania; 2Clinical Hospital of Orthopedics, Traumatology, and Osteoarticular TB “Foisor” Bucharest, B-dul Ferdinand 35–37, Sector 2, 021382 Bucharest, Romania; raresmircea@gmail.com

**Keywords:** *Fasciola hepatica*, human hepatic fasciolosis, pseudotumor, histopathologic diagnosis, case report, Romania

## Abstract

Human hepatic fasciolosis has been reported in 81 countries, some of which are endemic areas. In Europe, case reports from humans were published in Portugal, Spain, France, and Italy. Regarding Romania, we do not have any data on the prevalence of this parasitosis, with the exception of two cases of twins that were born in Romania and diagnosed in the last 37 years in Italy after joining their mother that lived there. We report the case of a patient diagnosed in Romania with chronic fasciolosis, presented as a hepatic pseudotumor that was diagnosed during the histopathological examination of the hepatic lesion. The patient received oral treatment with triclabendazole, two doses of 10 milligrams (mg) per kilogram (kg) of body weight, given 12 h apart, with no side effects during or after the treatment. The evolution of the patient was favorable. In conclusion, even in areas free of human fasciolosis, the presence of an anemic syndrome especially in children, abdominal pain in the upper quadrants, associated or not with other digestive manifestations, even more so associated with eosinophilia in the acute phase, should be carefully evaluated for ruling out a parasitosis such as fasciolosis even in countries where this diagnosis seems unlikely.

## 1. Introduction

Human hepatic fasciolosis has been reported in 81 countries, some of which are endemic areas, such as Bolivia, Peru, Ecuador, Iran, Egypt, Turkey, China, Vietnam, Nepal, Pakistan, and Syria. In North America, confirmed cases are attributed to foreign workers or immigrants from endemic areas. In Europe, case reports from humans were published in Portugal, Spain, France, and Italy. In total, 2.6 million people are estimated to be infected with *Fasciola* spp. [1]. Regarding Romania, we do not have any data on the prevalence of this parasitosis, two cases of twins that were born in Romania, in a cattle farmer family, were diagnosed in the last 37 years in Italy after joining their mother that lived there [2]. We report the case of a patient diagnosed in Romania with chronic fasciolosis, presented as a hepatic pseudotumor that was diagnosed during the histopathological examination of the hepatic lesion.

## 2. Case Report

We describe the case of a 55-year-old Caucasian female, who lived as a child in a rural area where sheep farming was practiced. She presents herself for investigations two months ago into the general surgery department for diffuse abdominal pain without dyspepsia, disturbances of gastrointestinal transit, fever, or weight loss. 

From the laboratory and imaging studies that were performed the following changes were reported: a slight eosinophilia of 0.360 k/microL (reference value—0.03–0.27 k/microL), a C-reactive protein of 11.63 mg/L (ref. val. 0–5 mg/L), and an abdominal ultrasound that highlighted a liver tumor. From the point of the serum tumor biomarker (CA19-9, CEA, and alpha-fetoprotein) the reported values were within reference values. An image from the abdominal ultrasound study is shown in Figure 1. 

The imaging studies were completed by an abdominal computerized tomography scan revealed the following pathological changes: liver with hypodense area in segments II–III in comparison with normal liver parenchyma, with an irregularly bossed contour, maximum dimensions of 50 × 47 × 30 mm, with enhancement following contrast administration in the periphery and transient hyperattenuation in the adjacent parenchyma in the arterial phase of the examination and a progressive loading with contrast and the tendency to homogenization in the venous and parenchymal phases. Cholecyst with homogeneous content, thin walls, without detection by computerized tomography scan cholelithiasis. The computerized tomography scan appearance is suggestive of an atypical hemangioma. Images from the abdominal CT-scan study are shown in Figure 2.

An abdominal magnetic resonance imaging (MRI) was also performed, and the following results were reported: hepatic measurements within normal range, clearly inhomogeneous capsular outline due to the presence of a slightly hyperintense T2 and T2-FS (Fat saturation), hypointense T1 lesion, without any changes in in-phase and out-of-phase sequences, located peripherally in segments II and III, markedly restrictive in Diffusion-weighted imaging/apparent diffusion coefficient (DWI/ADC), with a slight signal drop on high B acquisition (B800) and average ADC values of approx. 1100 × 10^−6^ mm^2^/s (at the level of the area with the most intense ADC hyposignal). The lesion is imprecisely delimited, with an infiltrative pattern appearance, it involves both the capsule without its distortion and the medial contour of the third segment, where it does not show a plane of delimitation from the lesser curvature of the stomach on any of the pre-contrast and postcontrast images. The lesion presents a spontaneously inhomogeneous structure, with fine hypointense T2 discrete serpiginous images, that includes the distal branches of the bile and portal canaliculi without their distortion. Postcontrast sequences are suboptimal due to artifacts. Progressive contrast uptake from the portal phase, with quasi-complete homogenization in the late phase that becomes isointense compared to the rest of the parenchyma (fading), and in the arterial phase perilesional transient hepatic attenuation differences images are present—the described aspects suggest lesion with magnetic resonance characteristics of malignancy (with a fibrotic component, possibly infiltrative periductal cholangiocarcinoma), or less likely, atypical hemangioma. No other suspicious local lesions in the rest of the hepatic parenchyma. Without dilation of the intrahepatic bile ducts and of the extrahepatic bile ducts. Cholecyst with homogeneous content, thin walls, without detectable by magnetic resonance imaging cholelithiasis. Images from the MRI study are shown in Figure 3.

The decision to undergo a surgical intervention was made via a laparoscopic approach. Under ultrasound guidance, a tumor formation of approximately 2 cm is detected at the level of hepatic segment III. A laparoscopic liver segment III resection was performed.

The histopathological examination of the lesion reveals numerous giganto-epithelioid granulomas with a central area composed of a dense inflammatory infiltrate, predominantly represented by eosinophils, on the background of which Charcot–Leyden crystals, necrosis, and cellular detritus are identified. In the periphery, epithelioid histiocytes have been identified as palisade in the central area, with the presence of lymphocytes around the histiocytes. Kupffer cells hyperplasia is also associated with the inflammatory process. One of the granulomas contains eggs of the *F. hepatica*. On Trichome Masson staining, Charcot–Leyden crystals stand out better. Conclusion: Eosinophilic hepatic granulomas with massive central necrosis, with histological appearance of *F. hepatica*. Histopathological aspects are reported in Figure 4 and Figure 5.

The patient received oral treatment with triclabendazole, two doses of 10 milligrams (mg) per kilogram (kg) of body weight, given 12 h apart, with no side effects during or after the treatment. The favorable evolution of the patient was the result of an interdisciplinary collaboration between multiple specialists, who helped to take care of this case.

## 3. Discussion

Human fasciolosis is a zoonosis disease caused by the ingestion of food, water, and combinations of both *F. hepatica* and *F. gigantica* species, with a geographical distribution in 81 countries [1,3], being responsible for approximately 2.6 million cases, with endemic presence in some areas, *F. gigantica* being responsible for most cases diagnosed in Asia and Africa [4,5,6,7]. Gabrielli et al. report two cases of fasciolosis in two twins, that were born in Romania, starting from the hospitalization of one of the girls with severe anemia, suspected during investigations of a biliary parasitosis. During a retrograde cholangiopancreatography that was performed, the release of *F. hepatica* in the sampled material was observed. Genetic examination revealed remarkable differences from native specimens, excluding the infestation in Italy [2]. Areas with high prevalence are also described in Bolivia, Peru, Ecuador, Argentina [8,9,10], Iran, Egypt, Turkey, China and Vietnam, Nepal, Pakistan, Syria [11], and Saudi Arabia [3]. In Europe, cases have been reported in Spain, Portugal, France [12,13], and Italy [2,14], while in North America, cases are identified mainly in immigrants or workers from other geographic areas [14,15,16,17,18,19].

*F. hepatica* is a parasitic trematode of the class Trematoda, phylum Platyhelminthes, order Plagiorchiida, family Fasciolidae, which has snails as an intermediate host and ruminants (sheep, cattle, buffaloes, etc.) as being the typical definitive hosts. In terms of the morphological aspect, it is compared to pumpkin seeds, or leaves, with dimensions reaching up to 3–4 cm. As previously mentioned, two species of *Fasciola* are recognized as responsible for fasciolosis in humans, namely *F. hepatica* with the previously mentioned dimensions and *F. gigantica*, which can reach up to 75 mm. *F. hepatica* has a simple morphological structure, without circulatory or respiratory systems, with two suction cups (suckers) on the ventral side (mouth and acetabulum), a digestive system with a simple structure—where waste materials are discharged—the mouth, which ensures the circulation of food principles and gases. An excretory system (with excretory and osmoregulatory functions) and a nervous system, represented by a pair of nerve ganglia grouped in the anterior segment of the *F. hepatica* (on either side of the esophagus, connected by a nerve ring) and lateral nerve cords, located in the posterior segment. The outer surface of *F. hepatica* is called tegument, a syncytial epithelium made up of a scleroprotein that protects against the lytic action in the digestive system of the host but also has a role in restoring the surface plasma membrane as well as in the absorption of nutrients, some of which are protective in the digestive system of the host, such as taurine. Through certain skin components, such as the Teg antigen, the immune response of the host is reduced contributing to the survival of the *F. hepatica*, which manages to complete its life cycle with the appearance of the adult form [15]. The genome is about 1.3 Gb and is contained in 10 pairs of chromosomes, which is characterized by a special polymorphism, which explains the adaptability to the environment but also the vaccination difficulties that are encountered [16]. At a hepatic level, they feed from the bile ducts and capillaries, through the anterior (mouth) sucker, by extracting lymph, and biliary tissues, and also by the tegument. In the liver of the infected ruminants, *F. hepatica* is a hermaphrodite worm that multiplies both sexually and asexually. It lays eggs in the bile ducts, from where they are passed out through excrement and can be released into the freshwater environment, with the release of larvae. There are over 30 snail species shown to be intermediate hosts of *F. hepatica* particularly species in the genera *Radix*, and *Galba*. The distribution of the intermediate host represents the key to sustaining the parasite in endemic areas and its distribution in overlapped to the one of *F. hepatica*. Miracidia hatch from eggs and seek to penetrate the snail’s intermediate host. Snail infection generates large numbers of cercariae from a single miracidium. Free swimming cercariae disperse a few meters where they encyst into metacercariae, attach to aquatic plants, or reach the grass. The mammal (in particular, the ruminants that are the typical definitive hosts) acquires the infection by ingesting metacercariae from water or leafy vegetables. Metacercariae excyst after ingestion in the intestine and release the immature parasites that migrate through the intestine wall, the abdominal cavity, and the liver to reach the bile ducts where another life cycle begins. Humans accidentally become definitive hosts in unusual circumstances by consumption of improperly cooked liver that is infected and contains immature forms of the parasite, from drinking contaminated water and eating freshwater plants such as watercress [1,2,3,4,5,14,17,18,19,20,21,22]. The life cycle diagram of *Fasciola* is reported in Figure 6.

Saba et al. report clinical data from an observational study of 53 cases diagnosed in Antalya, Turkey, over a period of 5 years (1998–2003), more frequently in female patients (32 versus 21 cases in male), 28 of them being diagnosed in the acute phase, and 19 in the chronic one. There were 6 other cases in the latency stage, either serologically, starting from eosinophilia or by imaging studies that were requested by patients for different digestive manifestations [24].

The infection in humans, in the acute phase, can evolve as a prolonged febrile syndrome, difficult to diagnose in areas where fasciolosis is not endemic, and liver lesions are suggestive of subcapsular hematomas. Cases with dyspeptic manifestations (such as nausea, vomiting, abdominal pain in the upper right or epigastric quadrant), and arthralgia are also described; other manifestations that have been reported are: liver abscesses, cholangitis, and biliary obstructions. The most common symptoms that were reported by the patients were epigastric abdominal, and then in the right hypochondrium pain, and malaise. For chronic cases, the most frequently described symptoms after pain in the same location were nausea and fatigue. Other symptoms described were pruritus, coughing, diaphoresis, and weight loss. In the chronic phase in animals, fasciolosis is characterized by the appearance of fibrosis, cirrhosis, or inflammation of the bile ducts. Aspects related to anemia both in the acute and chronic phases are described, especially in children, with weight loss and malnutrition, and neurocognitive disorders. Eosinophilia is present in the acute phase of the disease; it may be absent in the chronic phase [25,26,27]. Specific antibodies (excretion–secretion antigens (ESA)), by ELISA detection, were positive in all cases, also eosinophilia was present in all patients with acute infections and only in 4 out of 19 cases with chronic infection, in the same study published by Saba et al. [24].

Other authors have reported cases of reactive hepatitis, subcapsular hemorrhages [28,29], or severe liver necrosis [30]. In total, 26 cases were described with ectopic localizations of fasciolosis, namely pancreatic (3 cases), intestinal and abdominal localization (6 cases), at the level of the peritoneal cavity one case, in the skin and subcutaneous tissue (5 cases) or lymph nodes—on the laterocervical region (2 cases), ocular localization (4 cases), brain involvement (3 cases), one case with pulmonary involvement, and dorsal spine localizations in one case. Additionally, 2 cases had simultaneous brain and eye involvement [31]. In endemic areas, other cases of ectopic fasciolosis that are not adequately diagnosed are possible and should be given greater importance; investigational imaging studies using different techniques can be very helpful in these circumstances.

The treatment of choice for patients over the age of 6 years old is with triclabendazole, a benzimidazole drug, active on the immature and adult parasite, which interferes with the β-tubulin polymerization balance, disrupts the skin of the parasite, and inhibits the protein synthesis of *F. hepatica* [32]. It is recommended to use one or two doses of 10 mg per kg of body weight, given 12 or 24 h apart. Nitazoxanide is considered a therapeutic alternative, in doses of 500 mg twice a day, for 7 days, but with a lower efficiency (40–60%) [33]. Other alternative drugs, such as praziquantel, albendazole, niclofolan, metronidazole, emetine, and dehydroemetine, are no longer recommended due to their ineffectiveness.

Castillo Contreras et al. report a similar case of hepatic pseudotumor in acute fascioliasis, the lesion is localized in the 6th and 7th liver segments. The patient received initial treatment for *F. hepatica* with nitazoxanide but it was discontinued due to oral intolerance, and later she received a single dose of 250 mg triclabendazole with a favorable outcome [34].

Except for the cases diagnosed in Italy, coming from Romania, we did not find in our literature review any case reports with human hepatic fasciolosis diagnosed in our country in the last 37 years. In terms of historical data, Neghina R. et al. report sporadic cases in Romania over time. The authors report between 1929 and 1986; 17 cases were described in Romania [35]. Unfortunately, we were unable to assess the original data. At the same time, the unusual way of diagnosis, starting from the surgical intervention for a liver tumor, with the histopathological diagnosis of the case is also unusual. The reported clinical picture certainly brings beneficial scientific information.

## 4. Conclusions

In conclusion, even in areas free of human fasciolosis, the presence of an anemic syndrome, especially in children, and abdominal pain in the upper quadrants, associated or not with other digestive manifestations, even more so associated with eosinophilia in the acute phase, should be carefully evaluated for ruling out a parasitosis such as fasciolosis even in countries where this diagnosis seems unlikely.

## Figures and Tables

**Figure 1 healthcare-11-01451-f001:**
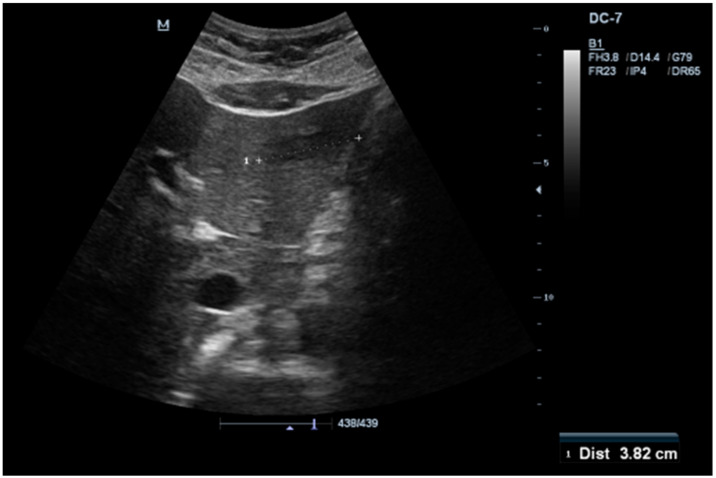
Abdominal ultrasound study highlighting the presence of a liver lesion of 3.82 cm in the 2nd and 3rd liver segments.

**Figure 2 healthcare-11-01451-f002:**
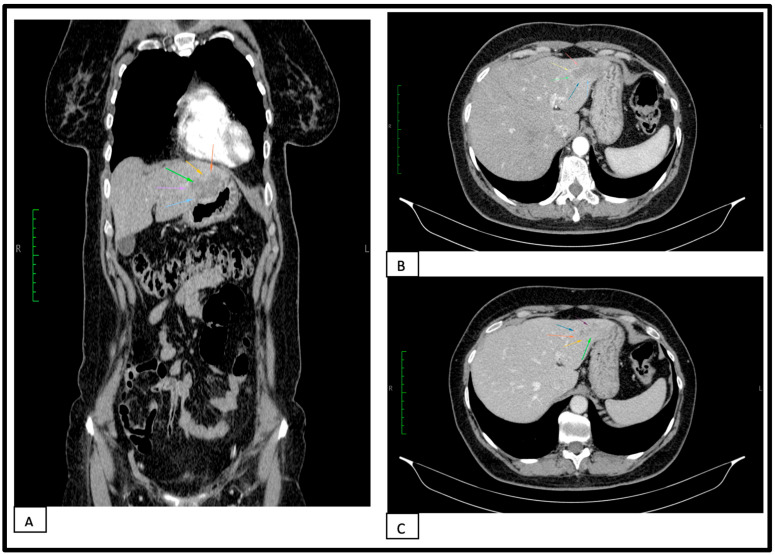
Abdominal CT-scan. (**A**) Early arterial phase in contrast-enhanced CT image in frontal plane image (colored arrows highlight the lesion). (**B**) Early arterial phase in contrast-enhanced CT image in transverse plane image (colored arrows highlight the lesion). (**C**) Late venous phase in contrast-enhanced CT image in transverse plane image (colored arrows highlight the lesion).

**Figure 3 healthcare-11-01451-f003:**
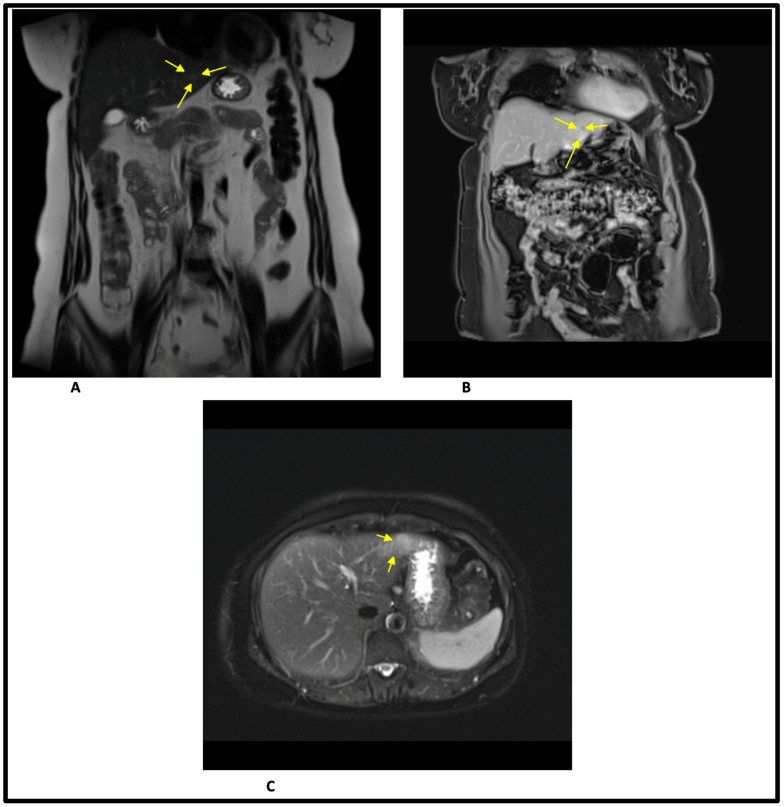
MRI scans of the abdomenum. (**A**) Single-shot T2-weighted MRI with deeplearning-based image reconstruction in frontal plane image (yellow arrows highlight the lesion). (**B**) T1-weighted -VIBE-Dixon image in frontal plane (yellow arrows highlight the lesion). (**C**) T2-weighted Turbo Spin Echo image in transvers plane (yellow arrows highlight the lesion).

**Figure 4 healthcare-11-01451-f004:**
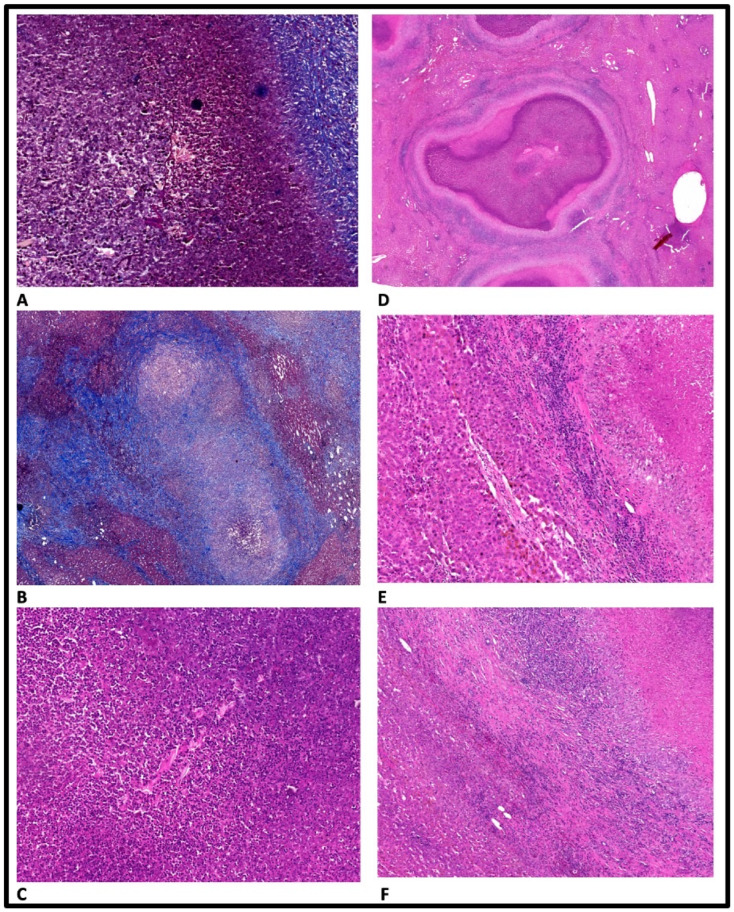
Histopathological aspects. (**A**)—Liver special staining, Charcot–Leyden crystals. (**B**)—Liver special staining, granuloma. (**C**)—Liver stained with hematoxylin and eosin, Charcot–Leyden crystals. (**D**)—Liver stained with hematoxylin and eosin, granuloma. (**E**)—Liver stained with hematoxylin and eosin, granuloma/liver tissue. (**F**)—Liver stained with hematoxylin and eosin, granuloma/liver tissue.

**Figure 5 healthcare-11-01451-f005:**
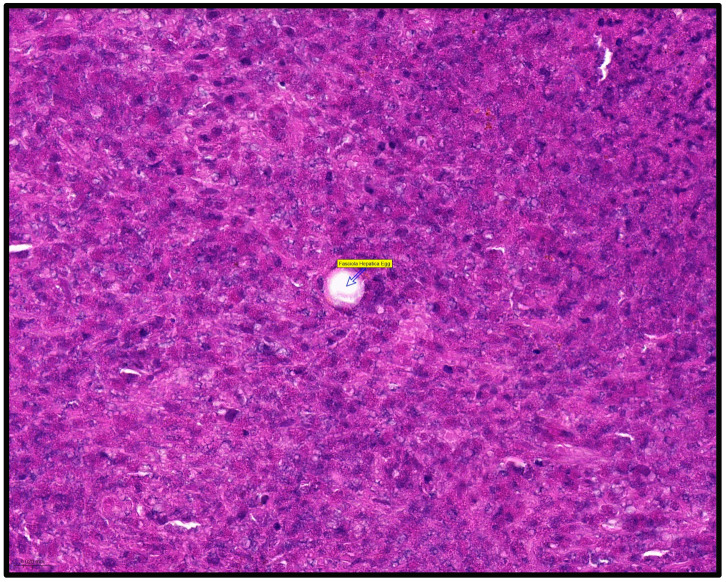
Histopathological aspects. Liver special staining, 50×, degenerated egg within the liver granuloma (blue arrow highlights the lesion).

**Figure 6 healthcare-11-01451-f006:**
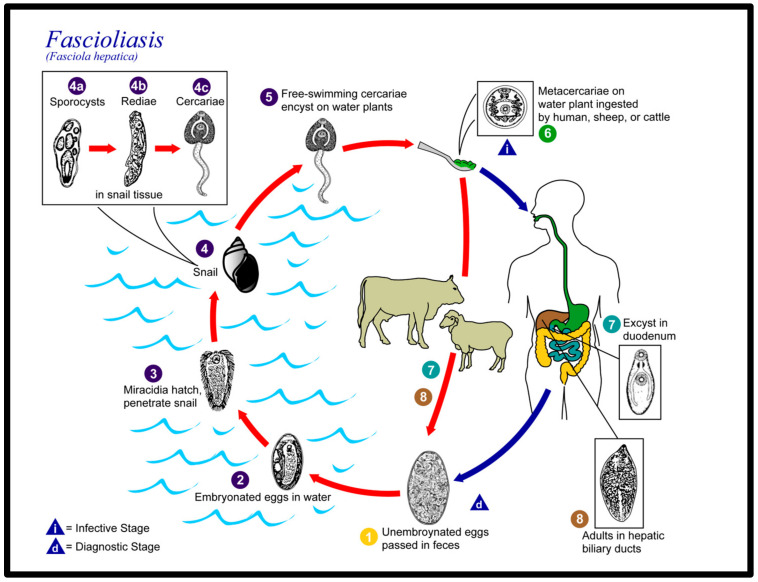
Life cycle diagram of *Fasciola*. CDC/Alexander J. da Silva, PhD; Melanie Moser. Reproduced from [23].

## Data Availability

All data generated or analyzed during this study are included in this published article.

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
