# Peer review of "The First Case of Human Hepatic Fasciolosis Presented as Hepatic Pseudotumor Histopathologically Diagnosed in Romania—A Case Report"

_healthcare, 2023, doi:10.3390/healthcare11101451_

Round 1

Reviewer 1 Report

In the present manuscript, the authors reported the case of a patient diagnosed in Romania with chronic fasciolosis, presented as a hepatic pseudotumor that was diagnosed during the histopathological examination of the hepatic lesion.

Some comments:

Lines 11 and 26: according to reference 1, delete “over”.

Lines 18 and 187: “in children” change to “especially in children”.

Line 27: delete “and”.

Line 28: delete “etc”.

Line 94: Figure 1 must be cited in the text.

Line 100: “by the ingestion of food most frequently infested with Fasciola hepatica and F. gigantica species”. Human fascioliasis infection sources include foods, water and combination of both. Delete “most frequently” and see: Mas-Coma S, Bargues MD, Valero MA. Human fascioliasis infection sources, their diversity, incidence factors, analytical methods and prevention measures. Parasitology. 2018 Nov;145(13):1665-1699. doi: 10.1017/S0031182018000914. Epub 2018 Jul 11. Erratum in: Parasitology. 2020 Apr;147(5):601. PMID: 29991363.

Lines 108-112: delete this paragraph (redundant information). These references should go in the introduction.

Line 116: Fasciola has a complex lifecycle that involves intermediate snail and definitive mammal hosts (not only ruminants), including humans. See also line 136.

Lines 129-130: “Through certain skin components…the survival of the F. hepatica”. Reference is needed.

Lines 131-134: “The genome is…vaccination difficulties that are encountered”. Reference is needed.

Line 135: “through the pharynges suction cup”.  F. hepatica use it mouth  sucker to pull off and suck up food, bile, lymph...

Line 139: delete “shell”. Please rewrite adequately and with a little more detail the life cycle of the parasite.

Line 143: “improperly cooked liver”. Humans can often acquire these infections through drinking contaminated water and eating freshwater plants such as watercress.

Lines 143-146: “Two species of Fasciola…Africa [1-5, 14-19]”. Move this information to the beginning of the discussion. 

Line 170: “Other authors have reported cases of reactive hepatitis, subcapsular hemorrhages, or severe liver necrosis”. References are needed.

Lines 171-175: Please revise the numbers and localizations of the ectopic cases described in reference 24.

Lines 179-181: “we did not find in our literature review any case reports with human hepatic fasciolosis diagnosed in our country”.

See: Neghina R, Neghina AM, Marincu I, Iacobiciu I. Epidemiology and history of human parasitic diseases in Romania. Parasitol Res. 2011 Jun;108(6):1333-46. doi: 10.1007/s00436-011-2256-0. Epub 2011 Feb 8. PMID: 21301873. 

Line 187: “the present changes certainly bring beneficial scientific information”. Please rephrase this sentence.

Once the species is named, use the abbreviated genus in subsequent mentions (lines 88, 101, 106…).

An article that could be consulted and cited by the authors:

Caravedo MA, Cabada MM. Human Fascioliasis: Current Epidemiological Status and Strategies for Diagnosis, Treatment, and Control. Res Rep Trop Med. 2020 Nov 26;11:149-158. doi: 10.2147/RRTM.S237461. PMID: 33273878; PMCID: PMC7705270.

Author Response

Sibiu, 21.04.2023

To

the Editors of Healthcare®

Dear Editor,

Dear Reviewer,

Thank you for reviewing our manuscript. Please find attached a revised version of our manuscript, “The first case of human hepatic fasciolosis presented as hepatic pseudotumor histopathologically diagnosed in Romania. A case report”.

Your and the reviewers’ comments were highly insightful and enabled us to greatly improve the quality of our manuscript. We have modified the manuscript in response to the comments. Attached is our point-by-point response to each comment.

Reviewer Comments:

Reviewer 1

In the present manuscript, the authors reported the case of a patient diagnosed in Romania with chronic fasciolosis, presented as a hepatic pseudotumor that was diagnosed during the histopathological examination of the hepatic lesion.

Answer: Thank you for taking your precious time to be able to assess our manuscript. The comments were highly insightful and enabled us to improve our manuscript.

Some comments:

Lines 11 and 26: according to reference 1, delete “over”. Lines 18 and 187: “in children” change to “especially in children”. Line 27: delete “and”. Line 28: delete “etc”.  Line 94: Figure 1 must be cited in the text.

A: Thank you for the suggestions. We performed the suggested changes.

Line 100: “by the ingestion of food most frequently infested with Fasciola hepatica and F. gigantica species”. Human fascioliasis infection sources include foods, water and combination of both. Delete “most frequently” and see: Mas-Coma S, Bargues MD, Valero MA. Human fascioliasis infection sources, their diversity, incidence factors, analytical methods and prevention measures. Parasitology. 2018 Nov;145(13):1665-1699. doi: 10.1017/S0031182018000914. Epub 2018 Jul 11. Erratum in: Parasitology. 2020 Apr;147(5):601. PMID: 29991363.

A: We performed the necessary change according to the reference.

Lines 108-112: delete this paragraph (redundant information). These references should go in the introduction.

A: We do not consider this information to be redundant, also in the introduction section of the manuscript already there is similar information.

Line 116: Fasciola has a complex lifecycle that involves intermediate snail and definitive mammal hosts (not only ruminants), including humans. See also line 136. “which has snails as an intermediate host and ruminants (sheep, cattle, buffaloes, etc.) as being the typical definitive hosts”

A: The complex lifecycle of Fasciola was rewritten. Thank you for the suggestion.

Lines 129-130: “Through certain skin components…the survival of the F. hepatica”. Reference is needed. Lines 131-134: “The genome is…vaccination difficulties that are encountered”. Reference is needed. Line 135: “through the pharynges suction cup”.  F. hepatica use it mouth  sucker to pull off and suck up food, bile, lymph...

A: References were provided

Line 139: delete “shell”. Please rewrite adequately and with a little more detail the life cycle of the parasite.

A: The complex lifecycle of Fasciola was rewritten. Thank you for the suggestion.

Line 143: “improperly cooked liver”. Humans can often acquire these infections through drinking contaminated water and eating freshwater plants such as watercress.

A: Thank you for the suggestion, our follow-up literature research highlighted this information also. We also included this information into the manuscript and regret that it was not included from the beginning.

Lines 143-146: “Two species of Fasciola…Africa [1-5, 14-19]”. Move this information to the beginning of the discussion.  Line 170: “Other authors have reported cases of reactive hepatitis, subcapsular hemorrhages, or severe liver necrosis”. References are needed. Lines 171-175: Please revise the numbers and localizations of the ectopic cases described in reference 24.

A: We addressed these issues as suggested.

Lines 179-181: “we did not find in our literature review any case reports with human hepatic fasciolosis diagnosed in our country”.

See: Neghina R, Neghina AM, Marincu I, Iacobiciu I. Epidemiology and history of human parasitic diseases in Romania. Parasitol Res. 2011 Jun;108(6):1333-46. doi: 10.1007/s00436-011-2256-0. Epub 2011 Feb 8. PMID: 21301873.

A: Thank you for pointing out this paper, we knew about it but unfortunately, we were not able to assess the reported cases between 1929-1986. We will report the data without being able to assess the original papers.

Line 187: “the present changes certainly bring beneficial scientific information”. Please rephrase this sentence. Once the species is named, use the abbreviated genus in subsequent mentions (lines 88, 101, 106…). An article that could be consulted and cited by the authors: Caravedo MA, Cabada MM. Human Fascioliasis: Current Epidemiological Status and Strategies for Diagnosis, Treatment, and Control. Res Rep Trop Med. 2020 Nov 26;11:149-158. doi: 10.2147/RRTM.S237461. PMID: 33273878; PMCID: PMC7705270. Done

A: We addressed these issues as suggested. We also took into consideration the suggested article

We hope that the revised form of the manuscript and our accompanying responses will be sufficient to make our manuscript suitable and accepted for publication in Healthcare®. We shall look forward to hearing from you at your earliest convenience.

With our best regards,

Sincerely yours,

Victoria Birlutiu, Prof. Habil. MD. PhD

Rares Mircea Birlutiu, MD PhD

Reviewer 2 Report

The authors present the description of the first case of human Fasciolosis in Romania. The case is well presented in terms of anamnesis and laboratory results. However, the authors only provide pictures of the histopathology. I would recommend to add the imaging studies pictures to provide more information and help other professionals who might encounter this parasitosis. Furthermore, I would recommend to add in the results section the handling of the patient (treatment) after the diagnosis and outcome of the intervention.

I would also recommend to check this sentence, present in both abstract and introduction, since it is grammatically incorrect:

 Regarding Romania, we do not have any data regarding the prevalence of this parasitosis, two cases of twins that were born in Romania, and diagnosed in Italy after joining their mother that lived there.

Finally, I would encourage the authors to provide more background about facioliosis diagnosis in the introduction and more information about the disease handling, treatment and outcome of previous reports in the discussion.

Author Response

Sibiu, 21.04.2023

To

the Editors of Healthcare®

Dear Editor,

Dear Reviewer,

Thank you for reviewing our manuscript. Please find attached a revised version of our manuscript, “The first case of human hepatic fasciolosis presented as hepatic pseudotumor histopathologically diagnosed in Romania. A case report”.

Your and the reviewers’ comments were highly insightful and enabled us to greatly improve the quality of our manuscript. We have modified the manuscript in response to the comments. Attached is our point-by-point response to each comment.

Reviewer Comments:

Reviewer 2

The authors present the description of the first case of human Fasciolosis in Romania. The case is well presented in terms of anamnesis and laboratory results. However, the authors only provide pictures of the histopathology. I would recommend to add the imaging studies pictures to provide more information and help other professionals who might encounter this parasitosis. Furthermore, I would recommend to add in the results section the handling of the patient (treatment) after the diagnosis and outcome of the intervention.

Answer: Thank you for taking your precious time to be able to assess our manuscript. The comments were highly insightful and enabled us to improve our manuscript. We were able to also provide ultrasound, CT scan, and MRI images from the preoperative studies. Also, data regarding the treatment of the patient and the outcome was included.

I would also recommend to check this sentence, present in both abstract and introduction, since it is grammatically incorrect:

Regarding Romania, we do not have any data regarding the prevalence of this parasitosis, two cases of twins that were born in Romania, and diagnosed in Italy after joining their mother that lived there.

Answer: We rephrased the sentence. Thank you for pointing this fact.

Finally, I would encourage the authors to provide more background about facioliosis diagnosis in the introduction and more information about the disease handling, treatment and outcome of previous reports in the discussion.

Answer: We also added more data into the introduction and discussion sections of the manuscript about the suggested topics.

We hope that the revised form of the manuscript and our accompanying responses will be sufficient to make our manuscript suitable and accepted for publication in Healthcare®. We shall look forward to hearing from you at your earliest convenience.

With our best regards,

Sincerely yours,

Victoria Birlutiu, Prof. Habil. MD. PhD

Rares Mircea Birlutiu, MD PhD

Reviewer 3 Report

The topic of the manuscript is interesting and useful, but some corrections are needed.

Lines 13-14. Regarding Romania, we do not have any data regarding the prevalence of this parasitosis, two cases of twins that were born in Romania, and diagnosed in Italy after joining their mother that lived there.” This sentence needs to be corrected. For example: Regarding Romania, we do not have any data on the prevalence of this parasitosis, with the exception of two cases …… Italy after joining their mother who lived there.

Lines 38-39. The sentence should be separated - We describe the case of a 55-year-old Caucasian female, who lived as a child in a rural area where sheep farming was practiced. She presents herself for investigations two months ago for diffuse abdominal pain without dyspepsia, disturbances of gastrointestinal transit, fever, or weight loss.

Line 42. In sentence “From the laboratory studies and paraclinical imaging studies that were performed. There is repetition in the sentence.

Line 44. ……..C-reactive protein of 11.63 mg/l (ref. val. -0-5 mg/l) … needs to be corrected.

Line 89-90. “One of the granulomas contains eggs of the Fasciola hepatica parasite Fasciola. There is no need to repeat „parasite Fasciola‘ in this sentence.

Line 89. Conclusion: Eosinophilic hepatic granulomas with massive central necrosis, with histological appearance of fasciolosis.“ Fasciola hepatica should be written instead of fasciolosis. Fasciolosis is a clinical diagnosis, but the histological test shows, for example, the body of the parasite or eggs.

Lines 150. “……….. 6 other cases in the latency stage, either serologically, starting from eosinophilia…….. “ The sentence should begin with words instead of a number. The same is in line 171 - 26 cases were described with ectopic localizations of fasciolosis.

Author Response

Sibiu, 21.04.2023

To

the Editors of Healthcare®

Dear Editor,

Dear Reviewer,

Thank you for reviewing our manuscript. Please find attached a revised version of our manuscript, “The first case of human hepatic fasciolosis presented as hepatic pseudotumor histopathologically diagnosed in Romania. A case report”.

Your and the reviewers’ comments were highly insightful and enabled us to greatly improve the quality of our manuscript. We have modified the manuscript in response to the comments. Attached is our point-by-point response to each comment.

Reviewer Comments:

Reviewer 3

The topic of the manuscript is interesting and useful, but some corrections are needed.

Answer: Thank you for taking your precious time to be able to assess our manuscript. The comments were highly insightful and enabled us to improve our manuscript.

Lines 13-14. “Regarding Romania, we do not have any data regarding the prevalence of this parasitosis, two cases of twins that were born in Romania, and diagnosed in Italy after joining their mother that lived there.” This sentence needs to be corrected. For example: Regarding Romania, we do not have any data on the prevalence of this parasitosis, with the exception of two cases …… Italy after joining their mother who lived there.

Answer: We addressed this raised issues and performed the suggested correction to be made. Thank you!

Lines 38-39. The sentence should be separated - We describe the case of a 55-year-old Caucasian female, who lived as a child in a rural area where sheep farming was practiced. She presents herself for investigations two months ago for diffuse abdominal pain without dyspepsia, disturbances of gastrointestinal transit, fever, or weight loss.

Answer: We addressed this raised issues and performed the suggested correction to be made. Thank you!

Line 42. In sentence “From the laboratory studies and paraclinical imaging studies that were performed”. There is repetition in the sentence.

Answer: We addressed this raised issues and changed the topic of the sentence. Thank you!

Line 44. “……..C-reactive protein of 11.63 mg/l (ref. val. -0-5 mg/l) … needs to be corrected.

Answer: We addressed this raised issues and performed a correction. Thank you!

Line 89-90. “One of the granulomas contains eggs of the Fasciola hepatica parasite Fasciola.” There is no need to repeat „parasite Fasciola‘ in this sentence.

Answer: We addressed this raised issues and performed the suggested correction to be made. Thank you!

Line 89. “Conclusion: Eosinophilic hepatic granulomas with massive central necrosis, with histological appearance of fasciolosis.“ Fasciola hepatica should be written instead of fasciolosis. Fasciolosis is a clinical diagnosis, but the histological test shows, for example, the body of the parasite or eggs.

Answer: We addressed this raised issues and performed the suggested correction to be made. Thank you!

Lines 150. “……….. 6 other cases in the latency stage, either serologically, starting from eosinophilia…….. “ The sentence should begin with words instead of a number. The same is in line 171 - 26 cases were described with ectopic localizations of fasciolosis.

Answer: We addressed this raised issues and performed the suggested correction to be made. Thank you!

We hope that the revised form of the manuscript and our accompanying responses will be sufficient to make our manuscript suitable and accepted for publication in Healthcare®. We shall look forward to hearing from you at your earliest convenience.

With our best regards,

Sincerely yours,

Victoria Birlutiu, Prof. Habil. MD. PhD

Rares Mircea Birlutiu, MD PhD

Reviewer 4 Report

The manuscript “healthcare-2307644; The first case of human hepatic fasciolosis presented as hepatic pseudotumor histopathologically diagnosed in Romania”  reported the first case of human hepatic fasciolosis presented as hepatic pseudotumor histopathologically diagnosed in Romania. Several major issues should be addressed:

Major comments:

1. What was the diagnosis before conducting the laparoscopic liver segment III resection?

2. The description of the pseudotumor during the laparoscopic liver segment III resection is needed.

3. The authors did not adhere to the CARE Guidelines and missed several sections. The authors need to provide a CARE checklist to make sure that all the information was provided in the manuscript. Include the checklist as a supplementary table (https://www.equator-network.org/reporting-guidelines/care/)

4.The authors missed to cite a similar case of pseudotumor hepatic fascioliasis (Acta Gastroenterologica LatinoamericanaVolume 43, Issue 1, Pages 53 - 582013)

5. In figure 1. A histological appearance of fasciolosis granulomas containing eggs of the Fasciola should be presented.

6, Treatment and outcome of the patients are needed.

Minor: 

-replace 50/47/30 mm by 50 x 47 x 30 mm (with x from insert).
-similar to 1100x10-6 mm2/s

Author Response

Sibiu, 21.04.2023

To

the Editors of Healthcare®

Dear Editor,

Dear Reviewer,

Thank you for reviewing our manuscript. Please find attached a revised version of our manuscript, “The first case of human hepatic fasciolosis presented as hepatic pseudotumor histopathologically diagnosed in Romania. A case report”.

Your and the reviewers’ comments were highly insightful and enabled us to greatly improve the quality of our manuscript. We have modified the manuscript in response to the comments. Attached is our point-by-point response to each comment.

Reviewer Comments:

Reviewer 4

  1. The manuscript “healthcare-2307644; The first case of human hepatic fasciolosis presented as hepatic pseudotumor histopathologically diagnosed in Romania”  reported the first case of human hepatic fasciolosis presented as hepatic pseudotumor histopathologically diagnosed in Romania. Several major issues should be addressed:

    Answer: Thank you for taking your precious time to be able to assess our manuscript. The comments were highly insightful and enabled us to improve our manuscript.

    Major comments:

    What was the diagnosis before conducting the laparoscopic liver segment III resection?

    Answer: Regarding your first questions - The diagnosis was “the described aspects suggest lesion with magnetic resonance characteristics of malignancy (with a fibrotic component, possibly infiltrative periductal cholangiocarcinoma), less likely atypical hemangioma”.

    The description of the pseudotumor during the laparoscopic liver segment III resection is needed.

    Answer: We provided data from the surgery. Than ypu for the suggestion.

    The authors did not adhere to the CARE Guidelines and missed several sections. The authors need to provide a CARE checklist to make sure that all the information was provided in the manuscript. Include the checklist as a supplementary table (https://www.equator-network.org/reporting-guidelines/care/) . 

    Answer: The CARE checklist was also added.

    The authors missed to cite a similar case of pseudotumor hepatic fascioliasis (Acta Gastroenterologica LatinoamericanaVolume 43, Issue 1, Pages 53 - 582013)

    Answer: Regarding the suggested manuscript - Thank you for the suggestion, probably we missed it due to the fact that it is in Spanish.

    In figure 1. A histological appearance of fasciolosis granulomas containing eggs of the Fasciola should be presented.

    Answer: Regarding figure no.1 - Unfortunately those are the images that were sent to us from the histopathological department together with the description that is included in the manuscript.

    Treatment and outcome of the patients are needed.

     Answer: The treatment and the outcome of the patients was also included.

    Minor: 

    -replace 50/47/30 mm by 50 x 47 x 30 mm (with x from insert).

    Answer: We also addressed these issues and performed the suggested corrections to be made.
    -similar to 1100x10-6 mm2/s

    Answer: We also addressed these issues and performed the suggested corrections to be made.

We hope that the revised form of the manuscript and our accompanying responses will be sufficient to make our manuscript suitable and accepted for publication in Healthcare®. We shall look forward to hearing from you at your earliest convenience.

With our best regards,

Sincerely yours,

Victoria Birlutiu, Prof. Habil. MD. PhD

Rares Mircea Birlutiu, MD PhD

Reviewer 5 Report

The first case of human hepatic fasciolosis presented as hepatic

pseudotumor histopathologically diagnosed in Romania

In this case report, it is discussed that a patient with chronic fasciolosis was first evaluated as hepatic pseudotumor and then diagnosed as chronic fasciolosis by histopathology. The subject of the study is very important in terms of the subject it deals with. Because, diseases caused by parasites (alveolar echinococcosis, visceral larva migrans, etc.) can be confused with other diseases. This study confirms that parasitic diseases should not be ignored and should always be kept in mind in the diagnosis of diseases.

Although the subject of the study is valuable, the manuscript is not well designed. Namely; In the case report part, the findings and evidence of fasciolosis are not clear. I couldn't see a parasitic structure in the figures. If parasitic structures (egg, tegument…) are seen in the histopathological sections, they should be shown in the figures. In addition, markers such as “arrow” should be used for what is desired to be shown in existing figures, how much magnification (40x, 100x…) is used in the figure description, and scales should be added to the figures.

If the parasitic structures are not proven by figures, how can we claim that this is due to fasciolosis? Therefore, this issue needs to be clarified first. Then minor corrections.

Author Response

To
the Editors of Healthcare®

Dear Editor-in-Chief,
Dear Editor,
Dear Reviewer,

Thank you for reviewing our manuscript. Please find attached a revised version of our manuscript, “The first case of human hepatic fasciolosis presented as hepatic pseudotumor histopathologically diagnosed in Romania. A case report”.

Regarding reviewer No. 5 comments that were sent after the decision for a major revision was sent to us.

Thank you for taking your precious time to be able to assess our manuscript.
We respect the comments and the raised issues. Regarding the histopathological examination of the specimens, it was done in a reference laboratory from Romania, and we fully believe in the result and based on that the patient received her treatment. As clinicians, our role is to treat patients and not to cause harm, and not to do experiments. Unfortunately, we are not pathologists so we would be able to assess the specimen by ourselves and we rely based on the received data.

Best regards,
The authors

Round 2

Reviewer 1 Report

Thank you for responding to my comments; however,  before I can recommend acceptance, there are still a few issues to address:

Once the species is named, use the abbreviated genus in subsequent mentions (lines 116, 118, 133) and use italics (line 177).

Lines 18 and 120: “patients” change to “patient”.

Lines 36-36: “Worldwide, it is estimated that approximately 2.6 million people are infested with Fasciola hepatica”. 2.6 million people are estimated to be infected with Fasciola spp.

Lines 184-186: Please rewrite this sentence because only in unusual circumstances do people become infected by eating undercooked liver (it must contain immature forms of the parasite).

Line 212 and 217: Please check that references 29 and 30 are placed correctly.

Line 228 and 235: Please check that references 32 and 33 are placed correctly.

Line 216: “brain involvement (3 cases)…. cerebral localization (3 cases)”. Please delete one of the repetitions.

Author Response

Sibiu, 05.05.2023

To

the Editors of Healthcare®

Dear Editor,

Dear Reviewer,

Thank you for reviewing our manuscript. Please find attached a revised version of our manuscript, “The first case of human hepatic fasciolosis presented as hepatic pseudotumor histopathologically diagnosed in Romania. A case report”.

Your and the reviewers’ comments were highly insightful and enabled us to greatly improve the quality of our manuscript. We have modified the manuscript in response to the comments. Attached is our point-by-point response to each comment.

Reviewer Comments:

Reviewer 1

Answer: Thank you for taking your precious time to be able to reassess our manuscript. The comments were highly insightful.

Thank you for responding to my comments; however,  before I can recommend acceptance, there are still a few issues to address:

Once the species is named, use the abbreviated genus in subsequent mentions (lines 116, 118, 133) and use italics (line 177).

A: Done. Thank you!

Lines 18 and 120: “patients” change to “patient”.

A: Done. Thank you!

Lines 36-36: “Worldwide, it is estimated that approximately 2.6 million people are infested with Fasciola hepatica”. 2.6 million people are estimated to be infected with Fasciola spp.

A: Thank you for the suggestion, we addressed the issue as you suggested.

Lines 184-186: Please rewrite this sentence because only in unusual circumstances do people become infected by eating undercooked liver (it must contain immature forms of the parasite).

A: We hope that the sentence is clear now.

Line 212 and 217: Please check that references 29 and 30 are placed correctly.

A: Thank you for pointing out this fact. We arranged the reference list.

Line 228 and 235: Please check that references 32 and 33 are placed correctly.

A: A: Thank you for pointing out this fact. We arranged the reference list.

Line 216: “brain involvement (3 cases)…. cerebral localization (3 cases)”. Please delete one of the repetitions.

A: Done!

We hope that the revised form of the manuscript and our accompanying responses will be sufficient to make our manuscript suitable and accepted for publication in Healthcare®. We shall look forward to hearing from you at your earliest convenience.

With our best regards,

Sincerely yours,

Victoria Birlutiu, Prof. Habil. MD. PhD

Rares Mircea Birlutiu, MD PhD

Round 3

Reviewer 1 Report

Thanks for responding to my comments. It is important that it is clearly established that the main ways people can become infected are by eating contaminated raw freshwater plants or by ingesting contaminated water. I suggest changing “permanent host” to “definitive hosts”. I have no more comments.

Author Response

Dear reviewer,

Thank you for the suggestion, we performed the change as suggested.